# Low-Pass Filtering Method for Poisson Data Time Series

**Victor Getmanov** [1,2,*], **Vladislav Chinkin** [1], **Roman Sidorov** [1,*], **Alexei Gvishiani** [1,2], **Mikhail Dobrovolsky** [1], **Anatoly Soloviev** [1,2], **Anna Dmitrieva** [1,3], **Anna Kovylyaeva** [1,3], **Nataliya Osetrova** [1,3] and **Igor Yashin** [1,3]

1. Geophysical Center of the Russian Academy of Sciences (GC RAS), 119296 Moscow, Russia; v.chinkin@gcras.ru (V.C.); adg@gcras.ru (A.G.); m.dobrovolsky@gcras.ru (M.D.); a.soloviev@gcras.ru (A.S.); andmitriyeva@mephi.ru (A.D.); aakovylyaeva@mephi.ru (A.K.); nvosetrova@mephi.ru (N.O.); iiyashin@mephi.ru (I.Y.)
2. Schmidt Institute of Physics of the Earth of the Russian Academy of Sciences (IPE RAS), 123242 Moscow, Russia
3. Scientific & Educational Centre NEVOD, National Research Nuclear University MEPhI (NRNU MEPhI), 115409 Moscow, Russia
* Correspondence: v.getmanov@gcras.ru (V.G.); r.sidorov@gcras.ru (R.S.)

**Abstract:** Problems of digital processing of Poisson-distributed data time series from various counters of radiation particles, photons, slow neutrons etc. are relevant for experimental physics and measuring technology. A low-pass filtering method for normalized Poisson-distributed data time series is proposed. A digital quasi-Gaussian filter is designed, with a finite impulse response and non-negative weights. The quasi-Gaussian filter synthesis is implemented using the technology of stochastic global minimization and modification of the annealing simulation algorithm. The results of testing the filtering method and the quasi-Gaussian filter on model and experimental normalized Poisson data from the URAGAN muon hodoscope, that have confirmed their effectiveness, are presented.

**Keywords:** Poisson data; time series; quasi-Gaussian filter; digital filtering; optimization; global minimization; annealing simulation algorithm

## 1. Introduction

The article proposes a low-pass filtering method for Poisson-distributed data time series, based on a specially developed digital low-pass filter with finite impulse response (FIR filter), with gain equal to one at zero frequencies and non-negative weighting factors.

Here, low-pass filtering is applied in order to reduce noise in Poisson-distributed data to ensure the recognition of emerging fluctuations of mathematical expectations in them. Poisson-distributed, or Poisson data are found in various physical systems, for example, related to the heliosphere and magnetosphere of the Earth; the fluctuations of mathematical expectations of these data may contain information regarding the structures and characteristics of these systems.

A particular feature of the Poisson data origin is that they contain sufficient noises; it is known, for example, from [1] that their variance is numerically equal to mathematical expectation. Noise reduction in Poisson data can be achieved using common FIR filters [2,3], to which, within the framework of this article, we refer the filters based on commonly used windowing techniques, frequency sampling and inverse Fourier transforms [4,5]. However, there are a number of scientific and technical problems for which their application is not fully effective, for example, (1) recognition of small (in size and duration) mathematical expectation fluctuations in Poisson datasets; (2) digital processing of Poisson data with small mathematical expectation values.

Common FIR filters can potentially be used for the mentioned tasks, and their synthesis can be implemented according to given dimensions and cutoff frequencies. The synthesis procedures for common FIR filters are, in essence, the variants of approximation

procedures for the specified species frequency response (FR) types; the accuracy of the FR approximations depends on the specified dimensions for the synthesized filters. Obviously, at large dimensions, the accuracy of these approximations is high and the errors in the resulting cutoff frequencies are small. For the case of small dimensions, the approximation accuracy turns out to be low and, as a consequence, cutoff frequencies are realized with significant errors which prevent low-pass filtering. We can assume that the filtering procedure proposed here should be performed by filters with low dimensions and cutoff frequencies and with gain values equal to one in order to avoid mathematical expectation distortions, and with non-negative weight factors in order to provide non-negativity of filtering results taking into account the Poisson property of the data.

The indicated problem leads to the need to formulate the synthesis problem for a special digital low-pass FIR filter, which takes into account the requirements—restrictions on dimensionality, cutoff frequency, gain at zero frequencies, and weighting factors.

Here, a FIR filter is proposed, which is further denoted as a quasi-Gaussian filter, the frequency response of which is formed on the basis of approximating a Gaussian function and ensuring the implementation of the mentioned constraints conditions using a special optimization method.

Gaussian filters, the frequency response of which is implemented based on the approximation of the Gaussian function, are widely used in modern scientific and technical practice [6,7]. However, as a rule, the known variants of Gaussian filters with the approximation of the frequency response do not take into account the above-mentioned conditions (restrictions).

Problems of digital processing of Poisson data time series from muon counters in muon detectors and telescopes [8], counters of elementary particles of alpha-beta-gamma radiation, photon counters, slow neutrons, etc. [9], taking into account their specificity, are relevant for experimental physics. Digital processing of Poisson data, including the Gaussian filtering application, can be outside of experimental physics, for example, in medical technology for imaging blood vessels and tumor therapy with particle beams, in measuring technology for tribological studies of the surfaces of metal parts, in astronomy for gamma telescopes, in muon tomography for recognizing cavities in rocks, and building structures and many other applications.

One of the applications of the designed filter proposed here is the digital processing of the data from the URAGAN muon hodoscope (MH) designed by NRNU MEPhI [10,11]. The MH is a computerized measuring device that estimates the intensities of muon fluxes by counting the number of elementary particles—muons—registered in its detector for a set of solid angles with a set time step. Within the framework of this article, MH can be interpreted as a distributed set of muon counters, consisting of primary and secondary information converters.

From each primary MH transducer, the initial Poisson data—time series of random non-negative integers $N(Tk)$—the quantities of Poisson-distributed events recorded in a given solid angle at time intervals $(T(k-1), \; T(k-1) + T_{0k})$, $k = 1, 2, \ldots, k_0$, where $T = 1$ minute. Due to the features of the MH design, registration intervals $T_{0k}$ are random with a uniform distribution law in the range $T_{0\,\min} \leq T_{0k} \leq T_{0\,\max} < T$.

From each secondary MH transducer, the 1-minute-sampled normalized Poisson data $Y(Tk)$ are generated for a given solid angle by reducing to one second and calculating the averaged normalized Poisson data $Y(T_0 n)$ with an hourly discreteness according to the following relations:

$$Y(Tk) = N(Tk)/T_{0k}, \, Y(T_0 n) = \frac{1}{60} \sum_{k=1+60(n-1)}^{k=60n} Y(Tk), n = 1, 2, \ldots, T_0 = T \cdot 60. \quad (1)$$

Data resulting from (1) are produced for the whole set of solid angles; next, they are placed into time series of matrix MH data in the database [12].

## 2. Method

### *2.1. Quasi–Gaussian Digital Low-Pass Filter*

2.1.1. Statement of the Problem

One-dimensional FIR filter synthesized here is built according to the following difference equation:

$$X(T_0n) = \sum_{s=0}^{s_0} a_s Y(T_0(n-s)), n = 1, 2, \ldots, \tag{2}$$

where $r_0 = s_0 + 1$ is the FIR filter dimension, $a^T = (a_0, a_1, \ldots, a_{s_0})$ is a weight factors vector, $X(T_0n)$ is the output time series, $Y(T_0n)$ is the FIR filter input—the hourly normalized Poisson data time series from MH according to (2), which begins from the values $Y(T_0(1 - s_0))$, $Y(T_0(1 - s_0 + 1))$, $Y(T_0(1 - s_0 + 2))$, .... Transfer function (TF) $H(j\omega T_0, a)$ for filter (2) is defined as follows:

$$H(j\omega T_0, a) = \sum_{s=0}^{s_0} a_s e^{-j2\pi\omega T_0 s}. \tag{3}$$

Here $\omega$ is the TF frequency parameter. For (3), a normalized fequence is introduced, $w, \omega T_0 = w\pi, 0 \leq w \leq 1.0$, and its discrete values are calculated: $w_l$

$$dw = 1.0/L_0, \; w_l = dw(l-1), \; l = 1, \ldots, L, L = L_0 + 1. \tag{4}$$

The frequency response (FR) $H(w_l, a) = |H(jw_l, a)|$, considering (3), is the following:

$$H(w_l, a)^2 = H_1^2(w_l, a) + H_2^2(w_l, a),$$

$$H_1(w_l, a) = \sum_{s=0}^{s_0} a_s \cos(2\pi w_l s),$$

$$H_2(w_l, a) = \sum_{s=0}^{s_0} a_s \sin(2\pi w_l s) \tag{5}$$

for discrete normalized frequencies $w_l, \; l = 1, \ldots, L$ according to (4). The cutoff frequency $w_c$ for FR (5) is found based on the equality $|H(jw_A, a)|^2 = 0.5$.

For a low-frequency FIR filter synthesis, the FR of the prototype filter is used, based on a Gaussian function $H_{0g}(w, w_{c0})$

$$H_{0g}(w, w_{c0}) = \exp(-(w/w_{c0})^2). \tag{6}$$

2.1.2. Synthesis Requiements

The problem of synthesis of the supposed FIR filter is solved based on the approximation of the FR function $H_{0g}(w_l, w_c)$ (6) in discrete points $w_l, \; l = 1, \ldots, L$ with a FR function $H_g(w_l, 0)$ according to (5). A functional $S(H_{0g}, a, w_c)$ is formed:

$$S(H_{0g}, a, w_c) = \sum_{l=1}^{L} [(\sum_{s=0}^{s_0} a_s C_s(w_l))^2 + (\sum_{s=0}^{s_0} a_s S_s(w_l))^2 - H_{0g}^2(w_l, w_c)]^2. \tag{7}$$

Obviously, the FR (5) represents a function which is polyharmonic in frequency $w_l$. In case the prototype filter FR frequency derivative has discontinuities or is subject to strong alternations, e.g., if FR is a trapezoidal function, then the FR of the synthesized FIR filter, obtained based on approximation, will contain fluctuations due to the so-called Gibbs effect. Elimination and reduction of these fluctuations are usually achieved by choosing a suitable smooth prototype filter FR function. The smoothness requirement is largely satisfied by the Gaussian function (6).It should be noted that the Gaussian function is naturally suitable for the FR of a low-pass filter, since its values (6) practically differ from zero only in the region of low frequencies.

The requirements listed in the Introduction lead to formalized requirements:

a.   Ensuring that the gain at zero frequencies is equal to one:

$$1 = H(0, a) = \sum_{s=0}^{s_0} a_s, a \in A_1, \quad A_1 = \{a : (1 = \sum_{s=0}^{s_0} a_s)\}; \tag{8}$$

b.   Ensuring non-negativity of coefficients:

$$a \in A_0, \quad A_0 = \{a : (0 \le a_s, s = 0, 1, \ldots, s_0)\}; \tag{9}$$

For the synthesis procedure, it is assumed to set a small value $r_0$, based on the a priori known duration of fluctuations, and some small cutoff frequency value $w_c$ for a prototype filter. The quasi-Gaussian filter synthesis procedure, consisting of finding the optimal coefficients $a_s^\circ$, $s = 0, 1, \ldots, s_0$, taking into account the requirements **a,b**, Equations (8) and (9) the predefined $r_0$ and $w_c$, is performed on the basis of the approximation problem, which reduces to the implementation of conditional minimization:

$$a^\circ(w_c) = \arg\{\min_{a \in A_0, a \in A_1} S(H_{0g}, a, w_c)\}. \tag{10}$$

For a given small dimension $r_0$ of the synthesized quasi-Gaussian filter and a given small cutoff frequency $w_c$ for a prototype filter, the value for cutoff frequency to be found for a quasi-Gaussian filter is $w_{cg}$, and the filter FR for the frequencies $w_l$ is denoted as $H_g(w_l, w_{cg}, a^\circ)$, $l = 1, \ldots, L$.

The minimization of (10) could be performed based on modified direct zero-order optimization methods, taking into account the restrictions (8) and (9). However, because the (7) functional is multi-extremal, traditional modified direct methods, for example, using the coordinate descent method, the Hook–Jeeves method, the random descent method, etc. [13] do not provide successful minimization. The listed methods, as a rule, lead to "getting stuck" with search procedures in local minima.

### 2.2. Quasi–Gaussian Filter Synthesis Procedure

We can synthesize the quasi-Gaussian filter based on the technology of stochastic global minimization of the (7) functional with the constraints (8) and (9) using the optimization algorithm for annealing simulation [14,15]. To implement it, we will use the simulannealbnd.mat software module from the Matlab Global Optimization Toolbox [16].

Let us form a parallelepiped of constraints $\overline{A}_0$ of dimension $r_0$ with boundaries $\overline{a}_r$, $r = 1, \ldots, r_0$—$a \in \overline{A}_0$, $\overline{A}_0 = \{a : (0 \le a_r \le \overline{a}_r, r = 1, \ldots, r_0)\}$ and a new—with respect to (7)—functional $\overline{S}(H_{0g}, a)$ with a penalty term taking into account the constraint equality (8). Let us implement the global minimization of $\overline{S}(H_{0g}, a)$ taking into account $\overline{A}_0$ using [16].

Let us set the initial vectors for the first iteration $a_1(I) \in \overline{A}_0$, uniformly distributed in $\overline{A}_0$, $I$—a single descent procedure , $I = 1, 2, \ldots, I_0$, $I_0$—a total number of descent procedures. Let us assume that each descent procedure consists of $m_0$—a total number of iterations, $m$—a single iteration, $m = 1, 2, \ldots, m_0$. During descent, we assume that the initial value of the vector of parameters for $(m + 1)$-st iteration is equal to the calculated optimal value for the vector of parameters for $m$-th iteration—$a_{m+1}(I) = a_m^\circ(I)$ . In each iteration, we perform $n_0$ descent steps, $n$ is a descent step number, $n = 1, 2, \ldots, n_0$. Next, we will calculate the sequence of the functional $S(m_0, I) = \overline{S}(H_{0g}, a^\circ)$ values and the corresponding optimal vectors $a^\circ(m_0, I)$, $I = 1, 2, \ldots, I_0$. For global minimization, we search for the optimal index $I^\circ$ corresponding to the minimum of the $S(m_0, I)$ functional, and the optimal vector $a^\circ$ using brute force:

$$I^\circ = \arg(\min_{1 \le I \le I_0} S(m_0, I)\}, a^\circ = a^\circ(m_0, I^\circ).$$

On Figure 1, the example plots of the minimized $S(m, I)$ functionals are displayed, depending on iteration number $m$ and the descent procedure number $I$. Functionals are

shown starting with $m = 2$, since for $m = 1$ their values are very large. Here, $m_0 = 20$; as the iteration number increases, the values of the functionals decrease. During the optimization process, a movement is made in a $r_0$-dimensional space from one local minimum to another.

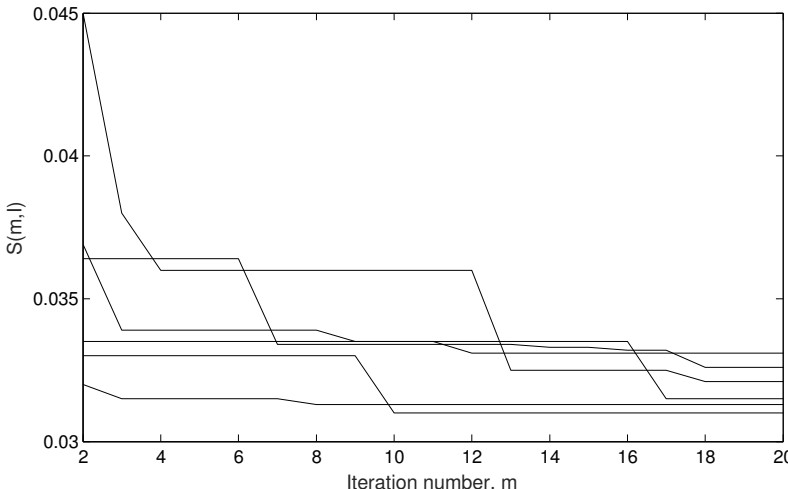

**Figure 1.** Plots of descent procedures—minimization of functionals $S(m, I)$, $I = 1, 2, \ldots, I_0$, $m = 1, 2, \ldots, m_0$.

Let us consider an example of quasi-Gaussian FIR filter synthesis. Based on the analysis of hourly experimental MH data from [12], it was found that the durations of possible fluctuations of the mathematical expectation in them were, on average, $\approx 10 \div 20$ h and more. The dimension value $r_0$, that could possibly allow the recognition of such fluctuations in mathematical expectations, was equal to 8. For a prototype filter FR (6), the parameter $w_{c0}$ was related to the assigned cutoff frequency $w_c$ based on (6)

$$(0.5)^{1/2} = \exp(-(w_c/w_{c0})^2), w_{c0} = w_c/(0.5 \cdot \ln 2)^{1/2}.$$

We assign the cutoff frequency $w_c = 0.1$, find $w_{c0}$ and define $H_{0g}(w, w_c)$—the prototype filter FR. By defining $L$ we set the number of discrete normalized frequencies $w_l$ of calculations of the functional (7) for $0 \leq w_l \leq 1.0$, let us assume that $L = 100$ in our calculations. The polyharmonic FR function $|H(jw)|$ (5) is formed from components performing $1, 2, \ldots, s_0$ fluctuations in this interval. For the accepted values $L$ and $r_0$, one period of the polyharmonic component with the maximum frequency corresponding to the number $s_0$ in (5), accounted for $\approx 15$ sampling points of normalized frequencies $w_l$, $l = 1, \ldots, L$, which fully provided a fairly accurate calculation of the functional (7) necessary for direct search.

Let us calculate the vector of factors $a^\circ$, form the synthesized quasi-Gaussian filter FR $H_g(w, w_{cg}, a^\circ)$ and define the cutoff frequency $w_{cg} = 0.175$ .

For the comparison, let us synthesize a common FIR filter using the fir1.mat module [3]. For the dimension $r_0 = 8$ and the assigned cutoff frequency $w_c = 0.1$ we find out the final cutoff frequency $w_{cf} = 0.275$; let us denote the FR as $H_f(w, w_{cf})$. On Figure 2, the FR plots for $H_{0g}(w, w_c)$, $H_g(w, w_{cg}, a^\circ)$, $H_f(w, w_{cf})$ are displayed. It is seen that, in case of low $r_0$ , the quasi-Gaussian filter FR was characterized by a better approximation to the prototype filter FR than the one of the common FIR filter.

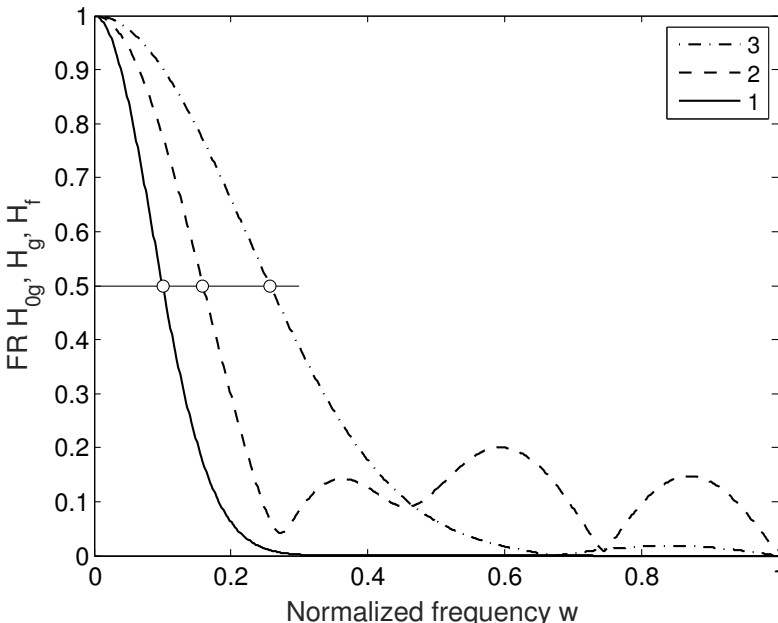

**Figure 2.** FR plots: $H_{0g}(w, w_c)$ (line 1), $H_g(w, w_{cg}, a°)$ (line 2), $H_f(w, w_{cf})$ (line 3).

Note that the proposed FIR filter, with the same dimension as the common FIR filter, made it possible to provide a lower value of the cutoff frequency than the realized cutoff frequency for the common FIR filter. The calculated cutoff frequencies of resulting FRs for common FIR filters synthesized using frequency sampling method and Fourier transforms [3,4] insignificantly (by ≈5–7% ) differ from the cutoff frequency $w_{cf} = 0.275$. This gives a reason to make a conclusion about the advantages of a quasi-Gaussian filter over standard FIR filters.

## 3. Results

### 3.1. Testing the Method and the Quasi–Gaussian Filter on Model Normalized Poisson Data

3.1.1. Testing on Model Hourly Data Using Statistical Modeling

Testing of the proposed method and quasi-Gaussian filter was carried out on model hourly normalized data using statistical modeling [17]. For this purpose, on the basis of the Matlab module exprnd.mat [18], exponentially distributed model random numbers $\tau_i$, $i = 1, 2, \ldots$ were generated, with their mean value $\tau_{M0}$ , and the evenly distributed random registration time intervals $T_{0k}, k = 1, 2, \ldots$ within the range $T_{0\ min} \leq T_{0k} \leq T_{0\ max}$. The number of Poisson model events $N_M(Tk)$ was counted on the registration time intervals $T_{0k}$. Finding $N_M(Tk)$ was carried out by solving the conditional maximization problems:

$$N_M(Tk) = \arg\{\max N_M\}_i > 0, \tag{11}$$

providing that $T_{0k} - \sum_{i=1}^{N_M} \tau$, where for $T_{0k}$, $k = 1, 2, \ldots k_0$ the range bounds $T_{0\ min} = 57$ s, $T_{0\ max} = 59.5$ s were assigned (see the Introduction section). Initial model 1-minute-sampled and normalized Poisson-distributed data were constructed according to (11) and the calculation of relations $\overline{N}_M(Tk)$, similar to (1):

$$\overline{N}_M(Tk) = N_M(Tk)/T_{0k}, k = 1, 2, \ldots k_0. \tag{12}$$

The modulation of the average number of Poisson events in order to model decreases (increases) in the mathematical expectation was carried out by specifying the mean value function $\tau_{M0}(Tk)$ on the intervals $(T(k-1), \ Tk)$ for $k$ from (12). For this, the relative modulation function $\mu(Tk)$, $k = 1, 2, \ldots, k_0$ was formed and the initial temporal index of the modulation decrease $k_a$, the duration of the decrease $dk_a$ and the depth of the relative

decrease $d\mu$ ;. The function $\mu(Tk)$ was represented by the relations $\mu(Tk) = 1 - d\mu$ for $k_a \leq k \leq k_a + dk_a$, $\mu(Tk) = 1$ for $1 \leq k < k_a$, $k_a + dk_a + 1 \leq k \leq k_0$.

For the calculation example, the average number of Poisson model events per minute was set $N_{M0} = 25$, normalized average $\overline{N}_{M0} = N_{M0}/T$, modulated normalized mean $\overline{N}_{M0}(Tk) = \overline{N}_{M0}\mu(Tk) = N_{M0}\mu(Tk)/T$, $k = 1, 2, \ldots, k_0$ and the parameter $\tau_{M0}(Tk) = 1/(\overline{N}_{M0}(Tk) - 1)$ was calculated.

Based on [18], random exponentially distributed numbers with $\tau_{M0}(Tk)$ and random evenly distributed values with $T_{0\ min} = 57s$, $T_{0\ max} = 59,5s$ were generated, with the use of which by (11), model Poisson data $N_M(Tk)$ and by (12)—normalized Poisson data $\overline{N}_M(Tk)$ were calculated. Further, similarly to (1), a time series of averaged model hourly normalized Poisson data was formed:

$$Y_M(T_0 n) = \frac{1}{60} \sum_{k=1+60(n-1)}^{k=60n} \overline{N}_M(Tk), n = 1, 2, \ldots, n_0, n_0 = ent(k_0/60). \qquad (13)$$

For modeling, we assumed $k_0 = 6000$, which corresponded to the model minute data produced during 4.166 days. For the modulation function, the values $k_a = 1920$, $dk_a = 1440$ and $d\mu = 0.02$ were taken. Model hourly averaged data $Y_M(T_0 n)$ for (13) with $n_0 = 100$, $n_{a1} = 32, n_{a2} = n_1 + dn_a\ dn_a = 24$.

Figure 3 shows an example of statistical modeling results: the jagged light gray line with index 1 displays the $Y_E(T_0 n)$ plot; the solid line with index 2 denotes the fragment of $X_{EG}(T_0 n)$ which is the result of filtering the model dataset using a quasi-Gaussian filter; for comparison, the dashed line with index 3 denotes the fragment $X_{EF}(T_0 n)$ which is the result of filtering the model dataset using the software module fir1.mat [3]. Model piecewise constant modulating function $Y_{M0}(T_0 n) = \overline{N}_M(T_0 n)$, represented by a dotted line (index 4), $m_0 + dm = Y_{M0}(T_0 n) = 0.4165$ for $1 \leq n < n_{a1}$, $n_{a2} \leq n < n_0$, $m_0 = Y_{M0}(T_0 n) = 0.4087$ for $n_{a1} \leq n \leq n_{a2}$, where the value of $dm = 0.833 \times 10^{-2}$ corresponded to the predefined 2% decrease. The plots show that the result of the quasi-Gaussian filter application (line 2) is a better approximation to the model piecewise constant modulation (line 4) than the result of a common FIR filtering (line 3).

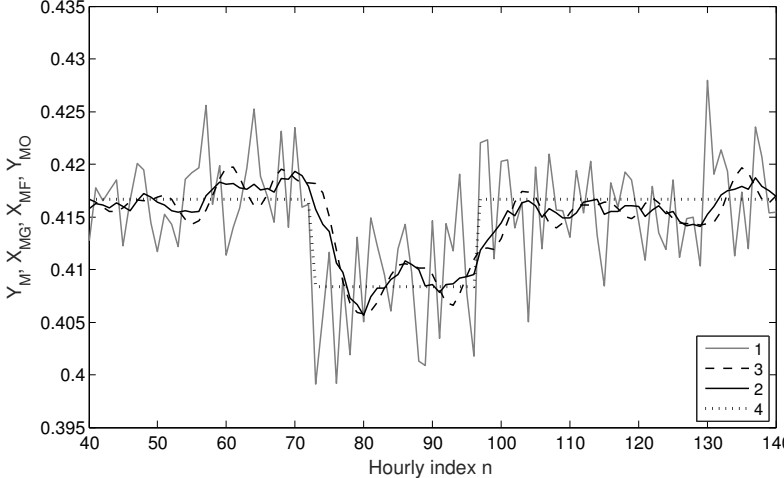

**Figure 3.** Fragments of model datasets $Y_M(T_0 n)$ (line 1), filtering results $X_{MG}(T_0 n)$ (line 2), $X_{MF}(T_0 n)$ (line 3) and model modulating function $Y_{M0}(T_0 n)$ (line 4).

The calculation of approximate estimates of filtering errors for the quasi-Gaussian filter and fir1-filter was performed by calculating the root-mean-square (RMS) errors according to the following formulas for datasets $Y_{M0}(T_0 n)$, $Y_{MG}(T_0 n)$, $Y_{MF}(T_0 n)$:

$$\sigma_{MG}^2 = \frac{1}{n_0} \sum_{n=1}^{n_0} (Y_{M0}(T_0 n) - Y_{MG}(T_0 n))^2, \sigma_{MF}^2 = \frac{1}{n_0} \sum_{n=1}^{n_0} (Y_{M0}(T_0 n) - Y_{MF}(T_0 n))^2. \qquad (14)$$

Results of a large number of tests performed for (14) showed that the $\sigma_{MG}$ error values for $X_{MG}(T_0 i)$ regarding $Y_{MG}(T_0 n)$ are, on average, 15–30% less than the corresponding $\sigma_{MF}$ error values for $X_{MF}(T_0 i)$. An overview of model $X_{MG}(T_0 i)$ and $X_{MF}(T_0 i)$ (Figure 3) made it possible to ensure that the minimum duration of the interval, within which recognition for the decrease $d\mu = 0.02$ can be performed, is 12–24 h.

The proposed method and the quasi-Gaussian filter provided more noise reduction than a common FIR filter. Consideration of the results of statistical modeling made it possible to draw a conclusion about the efficiency of the quasi-Gaussian filtering method.

### 3.1.2. Estimation of Mathematical Expectation and Its Root Mean Square Errors

Testing of the method and quasi-Gaussian filter for estimating the mathematical expectation and the RMS of its errors depending on $dn_a$—the duration of decreases and $d\mu$—the relative decrease value were carried out using statistical tests [17]. Random datasets $Y_M(s, T_0 i)$, $X_{MG}(s, T_0 i)$, $X_{MF}(s, T_0 i)$, $s = 1, 2, \ldots, M$, where $s$ is the number of the dataset, $M$ is the total quantity of datasets. The estimates of mathematical expectation $m_g^\circ(dn_a, d\mu)$ and RMS values $\sigma_g^\circ(dn_a, d\mu)$ for $X_{MG}(s, T_0 i)$ for a set of values $dn_a$ and $d\mu$

$$m_g^\circ(s, dn_a, d\mu) = \frac{1}{n_a} \sum_{n = n_{a_1}}^{n_{a_1} + dn_a} X_{MG}(s, T_0 n), m_g^\circ(dn_a, d\mu) = \frac{1}{M} \sum_{s = 1}^{M} m_g^\circ(s, dn_a, d\mu),$$

$$\sigma_g^\circ(s, dn_a, d\mu) = \frac{1}{n_a - 1} \sum_{n = n_a}^{n_a + dn_a} (X_{MG}(s, T_0 n) - m_g^\circ(s, dn_a, d\mu))^2, \quad (15)$$

$$\sigma_g^\circ(dn_a, d\mu) = \frac{1}{M} \sum_{s = 1}^{M} \sigma_g^\circ(s, dn_a, d\mu).$$

The coefficients of relative errors $\varepsilon_{gm}^\circ(dn_a, d\mu)$, $\varepsilon_{g\sigma}^\circ(dn_a, d\mu)$ of the quasi-Gaussian filter as ratios of errors $m_g^\circ(dn_a, d\mu) - m_0$ and RMS $\sigma_g^\circ(s, dn_a, d\mu)$ to the values of $dm$ reductions are the following:

$$\varepsilon_{gm}^\circ(dn_a, d\mu) = (m_g^\circ(dn_a, d\mu) - m_0)/dm, \varepsilon_{g\sigma}^\circ(dn_a, d\mu) = (\sigma_g^\circ(dn_a, d\mu))/dm \ldots \quad (16)$$

The coefficients $\varepsilon_{gm}^\circ$, $\varepsilon_{g\sigma}^\circ$, calculated for $dn_a, d\mu$, characterized the recognition capabilities of quasi-Gaussian filtering model decreases. Similarly, using (15) and (16) $m_f^\circ(dn_a, d\mu)$ and $\sigma_f^\circ(dn_a, d\mu)$ for $X_{MF}(s, T_0 n)$ and the coefficients $\varepsilon_{fm}^\circ(dn_a, d\mu)$, $\varepsilon_{f\sigma}^\circ(dn_a, d\mu)$. On Figure 4, the results of statistical tests are displayed, where $M = 500$. The $\varepsilon_{gm}^\circ(dn_a, d\mu)$ coefficients plots are the solid lines with indices 1, 2, and the $\varepsilon_{fm}^\circ(dn_a, d\mu)$ plots are the dashed lines with indices 3, 4. The coefficients $\varepsilon_{gm}^\circ$, $\varepsilon_{fm}^\circ$ are given depending on the duration with the values $dn_a = 12, 24, 48, 72$ h and relative decreases in $d\mu$, taking the values of 0.01 (indices 1, 3) and 0.03 (indices 2, 4).

The effect of quasi-Gaussian filtering was determined based on the calculation of $\delta\varepsilon_{fg,m}^\circ$—the rates of errors with respect to the mathematical expectations:

$$\delta\varepsilon_{fg,m}^\circ(dn_a, d\mu) = (\varepsilon_{fm}^\circ(dn_a, d\mu) - \varepsilon_{gm}^\circ(dn_a, d\mu))/\varepsilon_{gm}^\circ(dn_a, d\mu) \quad (17)$$

The results of the $\delta\varepsilon_{fg,m}^\circ$ calculations according to (17) for some $d\mu$ and $dn_a$ values are:

1. $\delta\varepsilon_{fg,m}^\circ = 0.115$ (11.5%) for $dn_a = 24$ and $d\mu = 0.01$;
2. $\delta\varepsilon_{fg,m}^\circ = 0.196$ (19,6%) for $dn_a = 24$ and $d\mu = 0.03$.

Analysis of the error values showed that the $\varepsilon_{gm}^\circ$ rate values appeared to be about 10–30% lower than the $\varepsilon_{fm}^\circ$ values. The nature of the dependencies of the estimates of the error coefficients for the $\varepsilon_{g\sigma}^\circ$ and $\varepsilon_{f\sigma}^\circ$ root mean square values for the same $dn_a$ and $d\mu$ parameters is almost the same: the $\varepsilon_{g\sigma}^\circ$ are also $\approx$10–30% lower than the $\varepsilon_{f\sigma}^\circ$. This means that, for the recognition of decreases small in duration and magnitude, the use of a quasi-Gaussian filter is more preferable than the use of a common FIR filter.

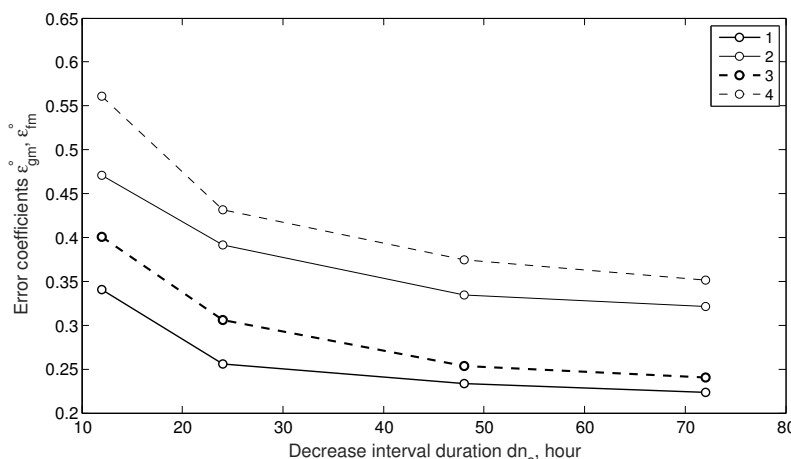

**Figure 4.** Results of calculating the coefficients of relative errors $\varepsilon_{gm}^{\circ}$, $\varepsilon_{fm}^{\circ}$.

*3.2. Testing the Method and the Quasi–Gaussian Filter on Experimental Normalized Poisson Data from the URAGAN Hodoscope*

Testing in this section consisted of determining the performance and capabilities of the proposed method and the quasi-Gaussian filter for recognizing small in duration and magnitude decreases in time intervals for the experimental hourly normalized Poisson data registered by the URAGAN hodoscope, taken from [12].

For analysis, a time interval was selected from 09/02/2017, 20:00 UTC to 09/18/2017, 15:00 UTC, with a total duration of 15.6 days. During this interval, the heliosphere was turbulent due to strong solar coronal mass ejections (CMEs) The CMEs that occurred on that period, caused intense geomagnetic storms that were discussed, for example, in [19,20]. The emerging CMEs caused modulations of muon fluxes recorded in MH and led to lower mathematical expectations (including the ones due to Forbush decreases) in Poisson MH data.

MH data were the matrix series of distribution functions of the intensities of muon fluxes $Y_E(i, j, T_0 n)$, defined in a rectangular region $i = 1, \ldots, N_1, j = 1, \ldots, N_2, N_1 = 90, N_2 = 76,$ $n = 1, 2, \ldots$. Solid angles correspond to azimuth and zenith indices $i, j$, $\varphi_i = \Delta\varphi(i-1)$, $\vartheta_j = \Delta\vartheta(j-1)$, $\Delta\varphi = 1^\circ$, $\Delta\vartheta = 4^\circ$ in which the registered particles were counted. MH data $Y_E(j_0, i_0, T_0 n)$ were a time series with indices $j_0, i_0$; the considered interval was determined for $n_{E\min} \le n \le n_{E\max}$, $n_{E\min} = 5900$, $n_{E\max} = 6275$ (counting hours for [12] began from the first hour of 2017).

Figure 5 shows the results of quasi-Gaussian filtering and interval recognition with reductions in mathematical expectation. The original data $Y_E(T_0 n)$ for $j_0 = 30$, $i_0 = 31$ were denoted by light gray jagged lines (index 1). Fluctuations in data with a period of $\approx$24 h and an amplitude of $\approx$0.0037–0.0040 are due to the daily rotation of the MH with the Earth. Line with index 2 depicts the data $X_{EG}(T_0 n)$ filtered based on quasi-Gaussian filter. The recognized intervals of intensity decrease, intensity recovery and intensity mathematical expectation decrease were denoted by a piecewise linear spline-like dashed line $X_{ES}(T_0 n)$ (index 3). Analysis of intervals 5969–6043, 6127–6189 based on $X_{ES}(T_0 n)$ leads to a conclusion that the mathematical expectation values of decreases on them were $\Delta m_1 = 0.01$, $\Delta m_2 = 0.005$ for the relative decrease rates $d\mu_1 = 0.027$, $d\mu_2 = 0.020$. For $Y_E(T_0 n)$ and $X_{EG}(T_0 n)$, the mathematical expectations on these intervals were $m_{E1}^{\circ} = 0.3505$, $m_{E1}^{\circ} = 0.3510$, and $m_{EG1}^{\circ} = 0.490$, $m_{EG2}^{\circ} = 0.3520$, respectively, on average. The errors of the mathematical expectations estimates were $\Delta m^{\circ} = 0.0010 - -0.0015$, which is 10–30% from the mathematical expectation values obtained, and this led to successful recognition of decreases with the relative decrease rates of 0.02–0.03.

Testing on experimental MH data made it possible to draw a conclusion about the efficiency of the quasi-Gaussian filtering method and its satisfactory capabilities for recognizing small fluctuations of the mathematical expectations.

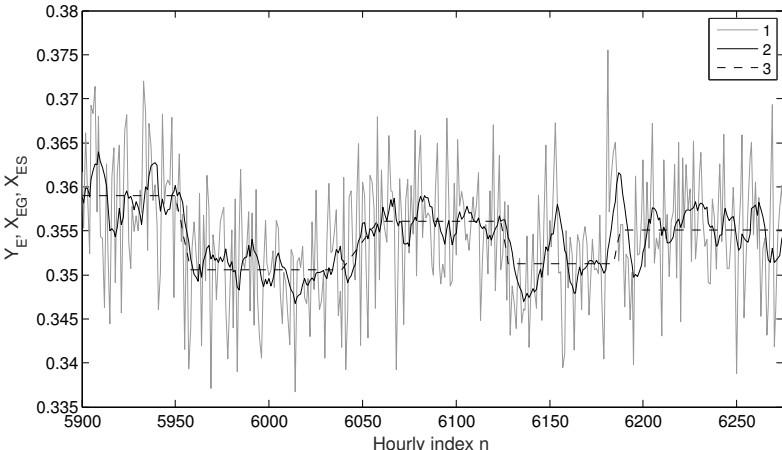

**Figure 5.** Results of quasi-Gaussian filtering and identification of regions with Forbush decreases: original data (line 1), filtered data (line 2), recognized intervals of various muon flux intensity (line 3).

## 4. Discussion

The comparison between the model data filtering result obtained using the proposed filter and the one obtained using the fir1 (plots on Figure 3) shows that the resulting time series are close to each other; however, the $X_{MG}(T_0 i)$ seems to be closer to the initial model. The main quantitative result of testing the method and the quasi-Gaussian filter on model normalized Poisson datasets included the calculations for (14) for a set of realizations/ The resulting errors $\sigma_{MG}$ for $X_{MG}(T_0 i)$, on average, by 15–30% less errors $\sigma_{MF}$ for $X_{MF}(T_0 i)$. This means that the proposed filtering method provided better filtering (noise reduction) than the standard FIR filter. Consideration of the results of statistical modeling made it possible to draw a conclusion about the efficiency of the method and the quasi-Gaussian filter.

Further tests of the new method on model data, aimed at estimating the mathematical expectation and its RMS errors with respect to the durations and magnitudes of model decreases, showed the method capabilities in disturbance recognitions. It can be seen on Figure 4 that the coefficients $\varepsilon_{gm}^\circ$ turned out to be less than the values of the coefficients $\varepsilon_{fm}^\circ$, on average, by about 10–30%. The nature of the plots of coefficients $\varepsilon_{g\sigma}^\circ$ and $\varepsilon_{f\sigma}^\circ$ for the RMS for the same parameters $dn_a, d\mu$ is almost the same—the coefficients $\varepsilon_{g\sigma}^\circ$ are less than the values of the coefficients $\varepsilon_{f\sigma}^\circ$, on average, also by $\approx$10–30%. From the point of view of recognizing decreases in duration and magnitude, the use of a quasi-Gaussian filter is more preferable than a common FIR filter.

Finally, tests made on real experimental datasets from a muon hodoscope display the method application to data processing and recognition of intervals of decreasing and recovering muon flux intensity. Due to the noise reduction in $X_{EG}(T_0 n)$, it became possible to clearly see the intervals of quiet data ( Figure 5), intervals with decreases and recoveries and intervals with declines in mathematical expectation; all these recognized intervals were denoted by a line $X_{ES}(T_0 n)$ (index 3 on Figure 5). On the intervals with the boundary points 5900–5954, 6057–6121, 6197–6276 there were quiet data, on the time intervals 5969–6043, 6127–6189 a decrease in mathematical expectation was observed, the time intervals 5955–5970, 6044–6056, 6122–6126, 6190–6196 corresponded to data with decreases and recoveries. On the intervals 5969–6043, 6127–6189, it is quite possible to recognize relative reductions in mathematical expectation. The errors of the mathematical expectations estimates were $\Delta m^\circ = 0.0010$–$0.0015$, which is 10–30% from the mathematical expectation values obtained, and this led to successful recognition of decreases with the relative decrease rates of 0.02–0.03 and an average duration of $\mathrm{approx}$ 10 h.

Testing the proposed method and quasi-Gaussian filter for data variants with indices $j_0 = 31$, $i_0 = 30$, allowed to obtain results that are almost similar to those depicted on Figure 5); the errors in the estimation of the boundary points of the sections during

recognition with depressions amounted to $\delta n \approx$ 2–5 h. Thus, testing on experimental MH data allowed us to make a conclusion about the efficiency of the method and the quasi-Gaussian filter and their satisfactory capabilities for recognizing mathematical expectation small in duration and magnitude.

## 5. Conclusions

The proposed filtering method for time series of normalized Poisson-distributed data, which was based on the developed digital low-pass quasi-Gaussian filter with a finite impulse response, a gain equal to one at low frequencies and non-negative weighting coefficients, turned out to be efficient; the FR of the low-frequency quasi-Gaussian filter of small dimension was characterized by a better approximation to the prototype filter FR than the FR of common FIR filters.

Testing the filtering method based on the quasi-Gaussian filter for the problems of recognizing small in duration and magnitude fluctuation decreases (increases) in mathematical expectations using statistical modeling and statistical tests have confirmed its effectiveness:

- The proposed method provided a decrease in errors in the filtered time series in comparison with the error values for standard FIR filters, by $\approx$15–30%; the method made it possible to recognize the mathematical expectation fluctuations with a relative decrease of 0.02 and duration of $\approx$12–24 h;
- The proposed method and the developed quasi-Gaussian filter provided relative error coefficients for mathematical expectation and root mean square values that appeared to be $\approx$10–30% less than the error coefficients for common FIR filters.

Testing the method and the low-frequency quasi-Gaussian filter on experimental Poisson data made it possible to draw a conclusion about its satisfactory capabilities for recognizing decreases with relative decrease coefficients $\approx$0.020–0.030.

The proposed method of noise reduction and a quasi-Gaussian filter have favorable prospects of using radiation particle counters for digital information processing in problems of experimental physics and measuring technology.

**Author Contributions:** Conceptualization, V.G., A.S. and I.Y.; methodology, V.G.; software, V.G. and V.C.; validation, M.D., A.G. and A.S.; formal analysis, V.C.; investigation, A.S.; resources, A.G.; data curation, A.D., A.K. and N.O.; writing—original draft preparation, V.G.; writing—review and editing, R.S. and M.D.; visualization, V.G.; supervision, I.Y.; project administration, A.S.; funding acquisition, A.G. All authors have read and agreed to the published version of the manuscript.

**Funding:** This work was funded by the Russian Science Foundation (project No. 17-17-01215).

**Data Availability Statement:** Data sharing is not applicable to this article.

**Acknowledgments:** The results of experiments presented in this research rely on data collected by the Scientific & Educational Centre NEVOD, National Research Nuclear University MEPhI. We acknowledge URAGAN muon hodoscope data provided by the NEVOD institution. This work employed facilities and data provided by the Shared Research Facility "Analytical Geomagnetic Data Center" of the Geophysical Center of RAS (http://ckp.gcras.ru/) (accessed on 19 April 2021). We would like to thank two anonymous reviewers whose valuable comments helped to improve the manuscript and properly demonstrate the results of our research.

**Conflicts of Interest:** The authors declare no conflict of interest.

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
