# Peer review of "Low-Pass Filtering Method for Poisson Data Time Series"

_applsci, doi:10.3390/app11104524_

Round 1

Reviewer 1 Report

The article is of interest to a wide range of readers and deserves publication. However, in my opinion, it needs some improvement. First of all, I believe that the introduction should present a broader scope of the developed approach. It is unlikely that the method of filtering data distributed according to Poisson applies only to the results obtained on the URAGAN muon Hodoscope. This distribution is quite common in a wide range of physical observations, and I believe that the authors would need to provide a broader overview of the possible areas of application of their proposed methodology. Otherwise, the reader immediately gets the impression of an extremely narrow field of application of this method, and the desire to continue reading the work disappears.
Further, the presented results also require a more detailed analysis. This applies to the results of processing and synthetic data and data obtained as a result of real field observations. Here, it is necessary to compare the general behavior of the curves obtained as a result of FIR and the proposed algorithm and give the relative deviation of the filtering results from each other and the true disturbance. Further, the authors write that the proposed method makes it possible to better preserve the spectral properties of the initial disturbance. Naturally, this raises a desire to compare the behavior of the spectra.
After completing such additional research, the article can be published in the journal Applied Sciences.

Author Response

Dear anonymous Reviewer,

Thank you for reviewing the manuscript of the article «A Method For Filtering Normalized Poisson Data From The URAGAN Muon Hodoscope» by V.Getmanov, V. Chinkin , R. Sidorov A. Gvishiani, M. Dobrovolsky, A.Soloviev, A.Dmitrieva, A.Kovylyaeva and I.Yashin. We agree with all the comments made. The new version of the article contains the necessary corrections and additions; the article has been substantially revised.

«The article is of interest to a wide range of readers and deserves publication. However, in my opinion, it needs some improvement».

In fact, there are 3 main issues to be improved:

  1. « First of all, I believe that the introduction should present a broader scope of the developed approach. It is unlikely that the method of filtering data distributed according to Poisson applies only to the results obtained on the URAGAN muon Hodoscope. This distribution is quite common in a wide range of physical observations, and I believe that the authors would need to provide a broader overview of the possible areas of application of their proposed methodology. Otherwise, the reader immediately gets the impression of an extremely narrow field of application of this method, and the desire to continue reading the work disappears».
  • In order for the results of the article to be focused on a wide range of applications of a sufficiently general nature, the title of the article has been changed - the mention of the muon hodoscope was removed from it and all the necessary information about it was placed at the end of the Introduction (lines 67-80) and in the Results section (subsection 3.2, lines 219-235). In addition, the introduction has been supplemented with a fairly broad overview of the applications of the proposed filtration method: (1) the options of particle counters are listed - the proposed method can be used for digital processing of information from them; (2) Some possible applications to which the developed approach can be applied are listed.

  1. «Further, the presented results also require a more detailed analysis. This applies to the results of processing and synthetic data and data obtained as a result of real field observations. Here, it is necessary to compare the general behavior of the curves obtained as a result of FIR and the proposed algorithm and give the relative deviation of the filtering results from each other and the true disturbance».
  • The presented results have been analyzed in detail. Comparison of the general behavior of the model curves obtained as a result of the operation of the quasi-Gaussian filter and the standard FIR filter relative to the original model data has been made (lines 188-195 in Subsubsection 3.1.1. in the Results section) Statements have been included, containing numerical characteristics on the deviation of filtering results from model data (the initial piecewise-constant modulating function).
  • In the Subsection 3.2, dedicated to testing the filtering method on experimental data, numerical characteristics comparing the deviations of the quasi-Gaussian filter curve with respect to the original experimental curve have been added (lines 236-251).

3. «Further, the authors write that the proposed method makes it possible to better preserve the spectral properties of the initial disturbance. Naturally, this raises a desire to compare the behavior of the spectra.»

 Unfortunately, there was some mistake or misunderstanding here. I could not find a place in the article to which this remark relates. Moreover, the article does not discuss spectra directly.

I ask the reviewer to review our reworked article again.  

On behalf of co-authors,

Best regards,

V.G. Getmanov

Reviewer 2 Report

To filter normalized Poisson data from the hodoscope, this paper proposed to use the digital quasi-Gaussian filter, and find that it is effective.

My main concern is the follows:

  • The Gaussian-filter is not new and there have already been much literature on Gaussian filter (if you search “Gaussian filter” on google scholar, you will see these previous studies). The paper should emphasize the novel point of your work, and distinguish yourself from previous literature.
  • There are numerous filters in the Matlab toolbox (e.g. Butterworth, Hanning, running average, Hamming), and you can design whatever filter that you want using the filter design toolbox in matlab. The authors could consider compare the results from these conventional filters with those from the Gaussian filter. Then they should state clearly in what aspects that the Gaussian filter is better.  Actually, I expect that other filters with appropriate parameters should do the job well as well.
  • Lines 254-269: you should present the evidence of these conclusions more clearly in Sections 3-4, probably with figures.
  • Line 259: Could you be more specific about standard FIR filters? After all there are numerous filters.

Author Response

Dear anonymous Reviewer,

Thank you for reviewing the manuscript of the article «A Method For Filtering Normalized Poisson Data From The URAGAN Muon Hodoscope» by V.Getmanov, V. Chinkin , R. Sidorov A. Gvishiani, M. Dobrovolsky, A.Soloviev, A.Dmitrieva, A.Kovylyaeva and I.Yashin, and for the comments that helped to improve the manuscript. We agree with all the comments made, and we have substantially reworked the manuscript according to your comments.

  1. «The Gaussian-filter is not new and there have already been much literature on Gaussian filter (if you search “Gaussian filter” on google scholar, you will see these previous studies). The paper should emphasize the novel point of your work, and distinguish yourself from previous literature»

In the Introduction of the revised article, the details have been added to emphasize the novelty of the work. The existing filtering tools do not solve the following problems: (1) widespread FIR filters based on the window method, frequency sampling and inverse Fourier transform for Poisson-distributed data time series with the indicated features are not fully effective; (2) the known types of Gaussian filters, in fact, are not focused on the implementation of restrictions on the sampling, cutoff frequency, gain and positiveness of the weight coefficients. The proposed quasi-Gaussian filter is designed to solve these problems. The corresponding statements, explaining the difference between the proposed quasi-Gaussian filter and conventional Gaussian filters with frequency response approximations, have been added in the Introduction section (lines 49-57).

  1. «There are numerous filters in the Matlab toolbox (e.g. Butterworth, Hanning, running average, Hamming), and you can design whatever filter that you want using the filter design toolbox in matlab. The authors could consider compare the results from these conventional filters with those from the Gaussian filter. Then they should state clearly in what aspects that the Gaussian filter is better. Actually, I expect that other filters with appropriate parameters should do the job well as well.»

We have updated the text of the Method section: the details regarding the comparison between the proposed quasi-Gaussian filter and common FIR filter types synthesized by the windowing method, frequency sampling and Fourier transform from the Matlab toolbox (lines 146-158). The result of the synthesis of standard FIR filters is that the obtained cutoff frequencies turn out to be much higher than the corresponding frequency of the quasi-Gaussian filter. This indicates that a quasi-Gaussian filter with lower errors implements low-pass filtering and is more preferable.

Modeling results (see their description, lines 173-195 in the Section 3.1.1) show that the quasi-Gaussian filter realizes the recognition of small mathematical expectation fluctuations in magnitude and duration in Poisson data.

3. «Lines 254-269: you should present the evidence of these conclusions more clearly in Sections 3-4, probably with figures.»

Additional numerical estimates of the results have been provided for Subections 3.1 and 3.2, At the end of Subsubsection 3.1.2, lines 205-216, an equation (17) for calculating relative error rates on mathematical models and numerical results based on calculations using this formula, confirming the conclusions on the errors, have been added. The quasi-Gaussian filtering errors are lower than the errors of filtering using common filters.

At the end of Subsection 3.2, numerical calculations have been added, related to experimental data, which confirm the effectiveness of the quasi-Gaussian filter for recognizing the fluctuations of the mathematical expectations in Poisson observations.

The corresponding details have been added to Section 4 and Section 5.

  1. «Line 259: Could you be more specific about standard FIR filters? After all there are numerous»

In the introduction, on page 1, lines 23-31, we have added an explanation of the term "standard (common) FIR filters", which is used in the article for the convenience of notation for FIR filters implemented by the windowing method, frequency sampling and Fourier transforms.

On behalf of the coauthors,

Best regards,

 V.G. Getmanov

Round 2

Reviewer 2 Report

The authors have addressed my concerns. Now I recommend publication.